# Comparative Gene-Expression Analysis of Alzheimer’s Disease Progression with Aging in Transgenic Mouse Model

**DOI:** 10.3390/ijms20051219

**Published:** 2019-03-11

**Authors:** Noman Bin Abid, Muhammad Imran Naseer, Myeong Ok Kim

**Affiliations:** 1Division of Life Science and Applied Life Science (BK 21), College of Natural Sciences, Gyeongsang National University, Jinju 660-701, Korea; noman_abid@gnu.ac.kr; 2Center of Excellence in Genomic Medicine Research, Department of Medical Laboratory Technology, Faculty of Applied Medical Sciences, King Abdulaziz University, Jeddah 21589, Saudi Arabia; mimrannaseer@yahoo.com

**Keywords:** Alzheimer’s disease, dementia, clinical diagnosis, gene expression

## Abstract

Alzheimer’s disease (AD) is a multifactorial neurodegenerative disorder characterized by progressive memory dysfunction and a decline in cognition. One of the biggest challenges to study the pathological process at a molecular level is that there is no simple, cost-effective, and comprehensive gene-expression analysis tool. The present study provides the most detailed (Reverse transcription polymerase chain reaction) RT-PCR-based gene-expression assay, encompassing important genes, based on the Kyoto Encyclopedia of Genes and Genomes (KEGG) disease pathway. This study analyzed age-dependent disease progression by focusing on pathological events such as the processing of the amyloid precursor protein, tau pathology, mitochondrial dysfunction, endoplasmic reticulum stress, disrupted calcium signaling, inflammation, and apoptosis. Messenger RNA was extracted from the cortex and hippocampal region of APP/PS1 transgenic mice. Samples were divided into three age groups, six-, nine-, and 12-month-old transgenic mice, and they were compared with normal C57BL/6J mice of respective age groups. Findings of this study provide the opportunity to design a simple, effective, and accurate clinical analysis tool that can not only provide deeper insight into the disease, but also act as a clinical diagnostic tool for its better diagnosis.

## 1. Introduction

Alzheimer’s disease (AD) is a multifactorial age-associated neurodegenerative disorder. AD gradually progresses by affecting different areas of the brain, including the cerebral cortex and hippocampus, starting from the frontal and temporal lobes and then gradually spreading to other parts of the brain [1]. Overproduction and accumulation of amyloid beta (Aβ) leads to the formation of Aβ plaques. Plaques are not only evident in extracellular spaces, but also when they start depositing on blood-vessel walls [2]. The second biggest hallmark of AD is aggregation of the microtubule-associated protein tau, which forms neurofibrillary tangles (NFTs) in neurons. Amyloid beta plaques and tau neurofibrillary tangles collectively lead to neurodegeneration, synaptic dysfunction, and dementia [3]. Clinical symptoms include severe and progressive memory loss, as well as a decline in language skills, including spatial and temporal orientations [4].

Recent epidemiological studies showed that Alzheimer’s disease and nonspecified dementia are more prevalent in women than in men [5]. According to the Australian Bureau of Statistics, approximately 66% of deaths due to dementia in Australian population were among women [6]. Recent evidence suggests that the prevalence of Alzheimer’s disease is consistent across the global population, and that the average duration of AD illness is 8–10 years, whereas the preclinical and prodromal stages span over two decades, suggesting that the mean onset of sporadic AD occurs at the age of 80 [7].

AD pathology is associated with the accumulation of Aβ plaques. Aβ is produced by the proteolytic cleavage of the amyloid precursor protein (APP) by gamma secretase and beta-secretase enzymes, including presenilin 1 (PSEN1) and presenilin 2 (PSEN2) [8]. To date, Aβ, APOE, and tau have been considered the main contributors to the onset of the disease, leading to neuropathological conditions such as selective neuronal death, synaptic loss, decline in neurotransmitters, and deposition of abnormal proteins, such as Aβ plaques and NFTs. Major risk factors for AD include diabetes mellitus, midlife hypertension, midlife obesity, a sedentary lifestyle, depression, smoking, and low educational attainment [9]. 

One of the biggest challenges related to AD is its diagnosis, not only during the prodromal stage but also in the dementia phase. Almost 35% of clinically diagnosed patients with AD were misdiagnosed and even showed negative Aβ positron emission tomography (PET) scans [10,11]. Age-associated diseases, as well as comorbidities including cardiovascular disease and hippocampal sclerosis, make this disease more complex to clearly diagnose. Previously, diagnosis was only possible by postmortem brain-sample analysis until the introduction of PET imaging using Pittsburgh compound B, which is a radioactive analog of fluorescent dye Thioflavin T that has the ability to cross the blood–brain barrier and interact with fibrillary Aβ [12]. Along with PET scans for the detection of amyloid beta concentration in cerebrospinal fluid, plasma and peripheral tissue have also been monitored to design more comprehensive diagnostic tools. 

Among the many challenges related to AD, one is the lack of detailed and comprehensive diagnostic tools [4]. To comprehend this multifactorial disease involving several molecular events spanning many genes, a detailed analytical tool is crucial. Gene-expression analysis is the most appropriate tool to address this issue. Quantification of gene expression has been performed using RNA-based techniques such as real-time quantitative PCR (Polymerase chain reaction), microarray analysis, and next-generation sequencing (NGS). These tools are very efficient and accurate but costly, making them unaffordable as diagnostic tools in the public sector of developing countries. The main purpose of the present study is to devise a cost-effective, efficient, and comprehensive method to dissect the molecular basis of AD. The outcome of the present study helps in pinpointing the most important genes involved in the onset and pathology of this disease, which could help in designing a low-cost but effective diagnostic assay for the clinical diagnosis of this challenging neurodegenerative disease.

Several pathological events at the molecular level collectively form an interlinked complex pathway that spans numerous genes involved in disease onset and progression. Detailed analysis of such a pathway can provide effective therapeutic strategies to target specific molecular events responsible for disease onset. In this regard, the Kyoto Encyclopedia of Genes and Genomes (KEGG) provides a detailed pathway that shows the roles of numerous genes involved in various molecular and pathological events associated with this disease [13]. The present study provides a detailed RNA-based gene-expression assay of 50 of the most important genes, covering all molecular events: APP processing, tau pathology, mitochondrial dysfunction, endoplasmic reticulum stress, neuroinflammation, and apoptosis. The present study analyzed in vivo mouse models for AD. The APP/Presenilin1 transgenic mouse was further divided into three age groups (six, nine, and 12 months) to visualize age-dependent changes in gene-expression levels. The outcome of this study provides insight into understanding the disease’s onset by dissecting its molecular events, and is also helpful for designing a diagnostic tool. This study also helps evaluate the roles of therapeutic agents to mitigate this disease and its pathology.

## 2. Results

### 2.1. Amyloid Precursor Protein Processing

APP disturbances lead to the synthesis and accumulation of amyloid beta, which is a major hallmark of neurodegeneration. This study includes several important genes involved in *APP* processing, such as *Adam10*, *App*, *Lpl*, *ApoE*, *Appbp1*, *Snca*, *Casp8*, *Psen1*, *Pen-2*, *Ide*, *Nep*, *Base1*, *Ncstn*, and *Aph1b*. PCR-based expression analysis showed that App, Snca, Appbp, Lpl, and Casp8 were significantly overexpressed in AD transgenic mouse models as compared to wild-type mice (Figure 1). Moreover, significantly decreased expression of *Adam10*, *ApoE*, *Psen1*, *Pen-2*, *Nep*, *Bace1*, *Ncstn*, and *Aph1b* was observed in AD groups. All of the above-mentioned genes are responsible for APP processing. Change in the expression level of these genes as compared to normal leads to amyloid beta production, which is responsible for senile plaques.

### 2.2. Tau Pathology

The tau protein, an abundant microtubule-stabilizing protein, has no direct involvement in AD pathology, but is instead an indirect consequence of Aβ overproduction. The hyperphosphorylation of tau protein is responsible for microtubule destabilization, thus leading to neurodegeneration and deficits in synaptic transmission and memory [14]. Genes including *Caln*, *Nos1*, *Gsk3b*, and *Mapt/tau* show significant overexpression in AD mice (Figure 2). Increased expression is evident with increasing age. Genes such as *P25* and *Cdk5*, which are involved in tau hyperphosphorylation, showed no overexpression in mice at six and nine months of age, while there was significantly increased expression at the age of 12 months. These expression analyses showed that tau pathology progressed with increasing age.

### 2.3. Endoplasmic Reticulum (ER) Stress

ER stress in postmortem brains from AD patients, animals, and in vitro models indicated that ER dysfunction might play an important role in causing AD pathogenesis [15]. ER stress is associated with many AD pathologies, including mutations in presenilin 1 and presenilin 2, Aβ production, tau pathology, and apoptosis. In this study, the expression of genes including *Ryr3*, *Ip3r*, *Psen1*, *Serca1*, *Perk*, *Ire1a*, *Atf6*, *Gq*, *and Plcb1* was analyzed (Figure 3). Expression analysis showed that *Ryr3*, *Ip3R*, *Ire1a*, and *Plcb1* were overexpressed in vivo in AD mice compared to the control group. By contrast, the expression levels of *Psen1*, *Serca1*, *Atf6*, and *Gq* were significantly lower in the disease group.

### 2.4. Calcium Signaling Disruption

Disruption of calcium signaling, induced by amyloid beta, is associated with memory dysfunction because a persistent increase in calcium level enhances long-term potentiation (LTP) and synaptic transmission [16]. The disruption of calcium signaling is considered a major aspect of disease pathology; thus, genes involved in calcium signaling were assessed in the present study. Genes controlling calcium signaling, including *Nmdar*, *Vdcc*, *Ryr3*, *Ip3r*, *Calm1*, *Erk/Mapk2*, and *Bad*, were analyzed (Figure 4a). The results of the expression analysis showed that the *Nmdar*, *Ryr3*, *Ip3r*, and *Mapk2* genes were overexpressed in AD groups, while the *Calm1* gene showed significant decrease in expression, and *Vdcc* and *Bad* showed no significant difference in expression in diseased brain samples compared to the wild type (Figure 4b).

### 2.5. Mitochondrial Dysfunction

Mitochondria are the most important organelle in eukaryotic cells, and are also known as the powerhouse of a cell because of their role in energy metabolism. In the brains of AD patients, mitochondrial function is disrupted because of cytosolic Aβ accumulation. Mitochondrial dysfunction is related to the association of Aβ with the Abad protein in mitochondria. Mitochondrial dysfunction leads to reactive oxygen species (ROS) and inflammation, which cause neurodegeneration and cell death. Genes that are involved in mitochondrial function, such as *Abad*, *Nmdar*, *Cx i–v*, and *Cycs*, were monitored in this study. The results showed that Abad and Nmdar were overexpressed in diseased samples (Figure 5), while *Cx iv* and *Cx v* showed significantly lower expression in diseased brain samples compared to wild-type samples. *Cx i* and *Cx ii* showed no significant change in expression in all age groups with AD as compared to normal control samples of respective age. While *Cx iii* showed no significant change in the six- and nine-month age groups, there was significant downregulation in the 12-month age group. (Figure 5). Detailed expression analysis showed that mitochondrial dysfunction worsens with aging. AD-affected mice suffered from energy deficits, and were thus faced with a decline in memory and cognitive abilities.

### 2.6. Inflammation and Apoptosis

Pathological processes linked to AD, such as Aβ plaques, tau pathology, endoplasmic reticulum stress, disruption in calcium signaling, and mitochondrial dysfunction, lead to the accumulation of reactive oxygen species and neuroinflammation, which leads to neuronal cell death. Inflammation and apoptosis are the final pathological events in the onset of AD and other neurodegenerative diseases. Neuronal cell death leads to memory impairment and cognitive decline. In the present study, we focused on inflammatory markers and genes responsible for neuroinflammation and cell death, such as *Tnf*, *Il1b*, *Casp12*, *Casp3*, *Casp7*, *Casp9*, *Cycs*, *Nos1*, and *Bid*, as shown in Figure 6.

Expression of genes related to neuroinflammation and apoptosis showed that amyloid-associated pathology leads to increased inflammation and cell death in transgenic mice, thereby leading to the onset of AD. Expression levels of all the genes included in this study are depicted collectively in the form of a heat map, as shown in Figure 7. Changes in expression patterns can be visualized in different age groups. *Tnf*, *Il1b*, and *Nos1*, which are markers of inflammation, showed significant overexpression. Similarly, *Casp 8* and *Casp 9* showed significant upregulation in all age groups. However, *Casp 3* and *Casp 7* only showed detectable upregulation in 12-month-old mice.

### 2.7. AD Onset with Age Progression

After the analysis of major molecular events involved in the onset of AD, overall analysis of the whole set of genes and their expression in three age groups showed that gene expression toward disease onset intensified with the increase of age. The gene heat map showed that most of the genes showed significant difference in expression at the nine-month age group, which was further aggravated as age progressed more (Figure 7a). The mean of gene expression in the three age groups showed that genes expression in the 12-month age group showed significant difference when compared to the mean of the six- and nine-month age groups (Figure 7b). Discrepancies in the normal function of biological processes occur whenever genes are either overexpressed or underexpressed as compared to normal conditions. The Venn diagram in Figure 7c shows that the six-month age group showed 21 overexpressed genes as compared to its control age group. Likewise, 28 and 29 genes were overexpressed in the nine- and 12-month age groups, respectively, as compared to their control counterparts, while 15 genes overlapped in all three age groups. Figure 7d shows the genes that were underexpressed as compared to control samples. The six-month age group showed 14 genes expressed at a lower level than the group samples, while 17 and 18 genes were underexpressed in the nine- and 12-month age group, respectively. Among these genes, about eight genes were common in all age groups (Appendix A).

## 3. Discussion

In this study, we performed gene-expression profiling of genes covering all major molecular events involved in the onset of AD. APP processing is one of the major molecular events and hallmarks of AD (Figure 8). This event involves several genes including *Adam10*, *App*, *Lpl*, *Apo-E*, *Appbp1*, *Snca*, *Casp8*, *Psen1*, *Pen-2*, *Ide*, *Nep/Mme*, *Bace1*, *Ncstn*, and *Aph1B*. Amyloid precursor protein, a transmembrane protein, is the precursor of amyloid beta. These transgenic mice with mutant APP genes showed APP overexpression. Our results showed significant difference in gene expression when compared with control mice. APP overexpression is the primary event in amyloid-beta-induced neurotoxicity and neurodegeneration. Adam10 is a member of the A Disintegrin and Metalloproteinase (ADAM) family that is involved in cleavage and plays a critical role in reducing the generation of Aβ peptides [17]. Reducing the expression of Adam10 produced improper APP cleavage and enhanced amyloid beta production. A significant decline in Adam10 expression is evident in the AD disease groups, and this expression further declines with aging. Presenilin protein (PSEN 1) is involved in APP cleavage and the production of amyloid beta fragments. Transgenic mice with mutant PS1 genes show abnormal expression compared to normal mice. This change in expression pattern causes the overproduction of amyloid beta, which leads to senile plaques. Further, among several risk factors for AD, the most impactful is the varepsilon4 isoform of apolipoprotein E (ApoE). A recent study highlights the role of isoform-dependent transcriptional regulation of APP by ApoE, explaining how ApoE enhances AD risk [18]. Decreases in the expression of Bace1, Pen2, and Nep are mostly related to APP cleavage and amyloid beta degradation, and they are responsible for amyloid-beta-related neurotoxicity (Andreoli et al., 2011) [19]. Overexpression of Appbp1 in primary neurons causes apoptosis, which leads to AD onset [20].

Tau pathology is among the major hallmarks of AD and dementia and, in this study, we evaluated genes involved in tau protein regulation and metabolism. Genes such as *Mapk/tau*, *Caln*, *P25*, *Nos1*, *Cdk5*, *and Gsk3β* were included in this study for expression analysis. Calcineurin is a calcium-dependent protein involved in the phosphorylation and regulation of the tau protein [21]. Overexpression of amyloid beta disrupts calcium signaling, as shown in our results based on the significant overexpression of Caln in AD mouse samples compared to control brain samples. The P25 protein is involved in the cleavage and accumulation of Cdk5, and P25 accumulation was observed and reported in AD patients [22,23]. The present study showed that there is no significant difference in the expression between AD and control mice at six months. This showed that overexpression of P25/Cdk5 was related to Aβ accumulation with increasing age. Nitric oxide synthase1 (Nos1) is responsible for generating nitric oxide, which leads to the nitrosylation of several proteins that are, in turn, associated with tau hyperphosphorylation [24,25]. Our results showed significantly increased expression of Nos1 in diseased samples. Glycogen synthase kinase 3 (Gsk3) is a serine/threonine kinase responsible for AD pathogenesis [26,27]. Gsk3 hyperphosphorylates tau protein at serine and threonine residues, leading to neuronal cell death.

Calcium signaling is an important homeostatic process in the brain. Dysregulation of calcium disrupts homeostasis and leads to neurodegeneration. In this study, we focused on several important calcium-signaling genes, *Nmdar*, *Vdcc*, *Ryr3*, *Ip3r*, *Calm1*, *Snca*, *Erk/Mapk2*, and *Bad*. The *N*-methyl-d-aspartate receptor (Nmdar) is a glutamate receptor and ion channel protein found in neurons, and, when activated by glutamate, it allows positively charged ions to flow through the cell membrane [28]. The NMDA receptor is very important for controlling synaptic plasticity and memory function [29]. Overexpression of NMDA receptors, as shown in this study, is evident in dementia and AD [30]. Along with NMDA receptors, voltage-dependent calcium channels are major regulators of calcium in neurons. Calcium (Ca^2+^) disruption, accompanied by long-term inflammation, leads to neurodegeneration in age-associated diseases [31,32]. Our data demonstrated that diseased samples showed no significant change in Vdcc expression in brain samples from six-, nine-, and 12-month-old transgenic mice. By contrast, overexpression of Inositol 3 phosphate receptor (Ip3r), a calcium regulator for the endoplasmic reticulum, was reported in diseased samples [33]. Calmodulin 1 (a calcium-modulated protein) acts as a regulator of calcium signaling and is involved in the activation of several kinases and phosphatases. Recent studies showed that binding Aβ oligomers to Calm1 is responsible for calcium dysregulation and the activation of mitogen-activated protein kinase (MAPK), which is responsible for synaptic dysfunction and memory impairment [34,35].

Endoplasmic reticulum stress is evident in Alzheimer’s disease. The ER plays an important role in protein transport, calcium signaling, and homeostasis [36]. To study ER stress, we analyzed the expression of important ER-related genes like *Ryr3*, *IP3r*, *Psen1*, *Serca1*, *Perk*, *Ire1a*, *Atf6*, *Gq*, and *Plcn1*. Our analysis showed that ryanodine receptor 3 and inositol 3 phosphate receptor, which are calcium regulatory receptors, are upregulated in AD brain samples. By contrast, sarcoplasmic/endoplasmic reticulum ATPase was reported to be downregulated in age-matched AD samples, a phenomenon that was reported to decrease with aging [37,38]. Moreover, the present study demonstrated that an AD model showed the downregulation of protein kinase RNA-like endoplasmic reticulum kinase (PERK), activating transcription factor-6 and Gq, which are modulators of endoplasmic stress and unfolded protein response [39,40].

Mitochondrial dysregulation is a major contributor to the onset of neurodegenerative disease. Amyloid beta-binding alcohol dehydrogenase (ABAD) expression is upregulated in disease-associated samples [41]. Overexpression of the N-methyl-D-aspartate (NMDA) receptor enhances calcium influx into mitochondria and causes metabolic disruption in neurons. Expression analysis of samples from the AD model demonstrated that mitochondrial Complexes I and II showed no significant changes compared to the wild type, while Complex III only showed downregulation in the 12-month brain sample. By contrast, significant downregulation of Complexes IV and V was reported in samples from an aged AD brain that showed disruption in mitochondrial activity [42].

## 4. Materials and Methods

The experimental procedures were approved by the Research Ethics Committee of the Department of Applied Life Sciences at Gyeongsang National University, Republic of Korea. (Approval number# GNU-170117-M0002, January 17, 2017). All experiments were performed in accordance with guidelines and regulations of the aforementioned research ethics committee.

### 4.1. Animals

Congenic double-transgenic B6.Cg-Tg (APPswe, PSENdE9)85Dbo/Mmjax AD model mice were purchased from The Jackson Laboratory (Bar Harbor, ME, USA). These mice express a chimeric mouse–human amyloid precursor protein bearing the Swedish mutation (Mo/HuAPP695swe), and a mutant human Presenilin 1 protein (PS1-dE9) in neurons of the central nervous system. C57BL/6J (wild-type) mice were purchased from Samtako Bio (Osan, Korea). Mice were housed under a 12 h light/12 h dark cycle at 25 °C with ad libitum access to food and water. Mice were euthanized, and the hippocampus region of the brain was dissected out for RNA extraction.

### 4.2. RNA Isolation and cDNA Synthesis

The brain samples of transgenic (*n* = 3 per group) and wild-type mice (*n* = 3 per group) from 3 different age groups, 6, 9, and 12 months of age, were incubated at 65 °C for 10 min and cooled on ice. Subsequently, 4 μL 5× first-strand reaction buffer (375 mM KCl and 15 mM MgCl_2_ in 250 mM Tris-HCl), pH 8.3 (Invitrogen, Carlsbad, CA, USA), and 2 μL 100 mM DTT (Invitrogen) were added to the vials and, after incubation for 2 min at room temperature, 1 μL 200 U per μL SuperScriptTM III Reverse Transcriptase (Invitrogen) was added. After mixing contents by gentle agitation, vials were placed in a Gene Amp^®^ PCR system (Applied Biosystems, Foster City, CA, USA).

### 4.3. Gene-Expression Analysis

The expression of different genes involved in AD pathology was determined in the brain using end point RT-PCR on 6-, 9-, and 12-month-old transgenic and control mice of respective age groups 6, 9, and 12 months by using primers (Table 1). GAPDH was used as a reference gene. End-point RT-PCRs were completed using Amplitaq Gold 360 Master Mix (Thermo Fisher Scientific, Waltham, MA, USA) in a total reaction volume of 25 μL containing 5 μL cDNA and 0.1 μM primers against specific gene transcripts. All PCRs were performed in a linear amplification range over 25 amplification cycles comprising an initial denaturation step of 94 °C for 5 min, a core cycle comprising 94 °C for 45 s, 55 °C for 30 s, 72 °C for 45 s, followed by a final extension of 72 °C for 7 min. PCR products were separated by TAE gel electrophoresis, visualized by ethidium bromide staining and quantified by densitometry using a Gel Doc XR+ Molecular Imager System (Bio-Rad, Hercules, CA, USA).

### 4.4. Statistics and Analysis

Gene amplification was analyzed by using Quantity One software (Bio-Rad). PCR data were investigated by one-way ANOVA coupled with Dunnett’s test using Graph Pad Prism 6 software. Heat-map analysis was done by using the Orange Data Mining Toolbox in Python.

## 5. Conclusions

The present study provided an overview of this multifactorial disease but also dissected details of pathological events at the molecular level. This study may help in designing diagnostic assays that can be used to analyze the cerebrospinal fluid of AD patients. In recent years, several natural proteins have been used as therapeutic agents for treating AD. Osmotin is an emerging and promising plant protein that has shown neuroprotective effects and reduced amyloid beta toxicity in a transgenic mouse model and neuronal cells [43,44]. Osmotin has also been reported to attenuate neuroinflammation [45], as well as provide protection against ethanol-induced apoptotic neurodegeneration [46]. To explore the molecular mechanism of this kind of therapeutic agent, more-detailed assays are required. The present study provides a preliminary background to design a valid and effective diagnostic tool. Our future plans include a detailed study including data from human brains with Alzheimer’s disease, and the use of next-generation sequencing and microarrays to cover a broader gene spectrum. Once a whole set of genes is compiled, we aim to device an economical, effective, accurate, and reproducible analysis tool. Assays based on such analysis would provide an excellent opportunity for the diagnosis of challenging diseases like Alzheimer’s and Parkinson’s.

## Figures and Tables

**Figure 1 ijms-20-01219-f001:**
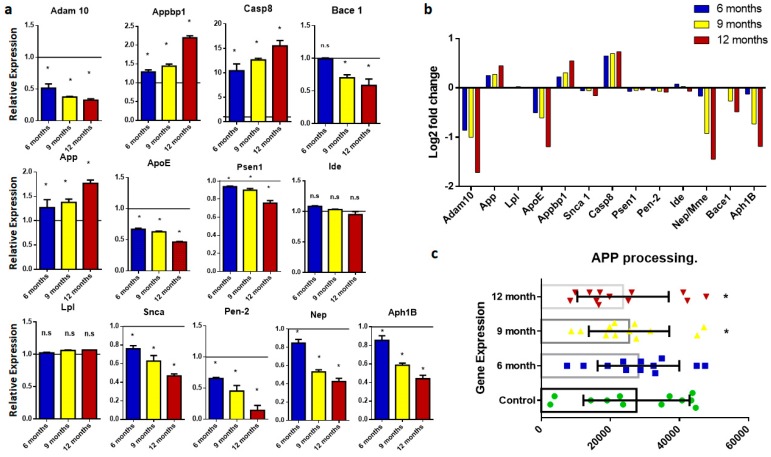
Expression analysis of genes involved in amyloid precursor protein procession. (**a**) Histograms representing relative mRNA expression of *Adam10*, *App*, *Lpl*, *Appb1*, *ApoE*, *Snca*, *Casp8*, *Psen1*, *Pen-2*, *Bace 1*, *Ide*, *Nep*, *Ncstn*, and *Aph1B*. Represented values are the mean and SD obtained from three individual experiments performed for each sample. The line on the x-axis at 1.0 shows the gene expression of control samples. (**b**) Bar plots representing fold-change values represented in the log2 scale showing gene expression in the brain of six-, nine-, and 12-month-old samples. Baseline shows expression in control samples. (**c**) Scattered-dot plot shows gene expression involved in amyloid protein processing. * Expression is significantly different to wild type (WT) (*p* ≤ 0.05). n.s., Nonsignificant.

**Figure 2 ijms-20-01219-f002:**
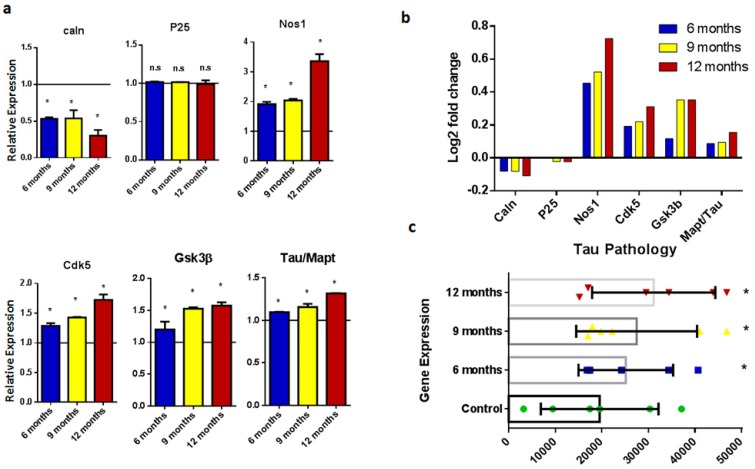
Expression analysis of genes involved in taupathology. (**a**) Histograms representing relative mRNA expression of *Caln*, *Cdk5*, *p25*, *Nos1*, *Gsk3b*, and *tau*. Represented values are the obtained mean and SD from three individual experiments performed for each sample. The line on the x-axis at 1.0 shows the gene expression of control samples. (**b**) Bar plots representing fold-change values represented in the log2 scale showing gene expression in the brain of six-, nine-, and 12-month-old samples. Baseline shows expression in control samples. (**c**) Scattered-dot plot shows gene expression involved in tau pathology. * Expression is significantly different to WT (*p* ≤ 0.05). n.s., Nonsignificant.

**Figure 3 ijms-20-01219-f003:**
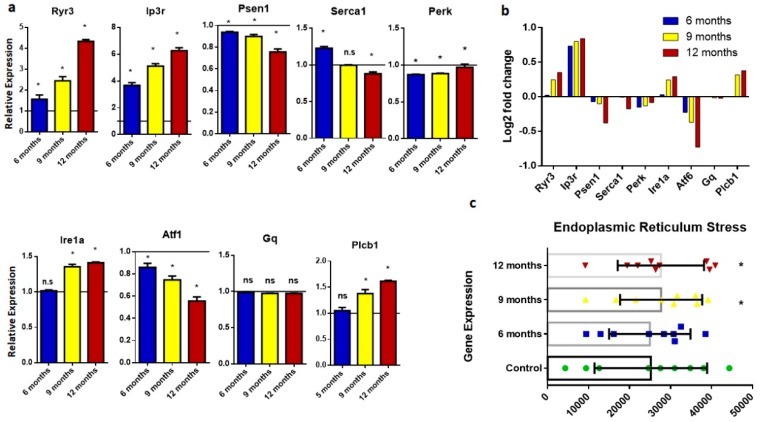
Expression analysis of genes involved in endoplasmic reticulum stress. (**a**) Histograms representing relative mRNA expression of *Ryr3*, *ire1a*, *Ip3r*, *atf1*, *Psen1*, *Gq*, *Serca1*, *Plcb1*, and *Perk*. Represented values are the mean and SD obtained from three individual experiments performed for each sample. The line on the x-axis at 1.0 represents the gene expression of control samples. (**b**) Bar plots representing fold-change values represented in the log2 scale showing gene expression in the brain of six-, nine-, and 12-month-old samples. Baseline shows expression in control samples. (**c**) Scattered-dot plot showing expression of genes involved in endoplasmic reticulum stress. * Expression is significantly different from WT (*p* ≤ 0.05). n.s., Nonsignificant.

**Figure 4 ijms-20-01219-f004:**
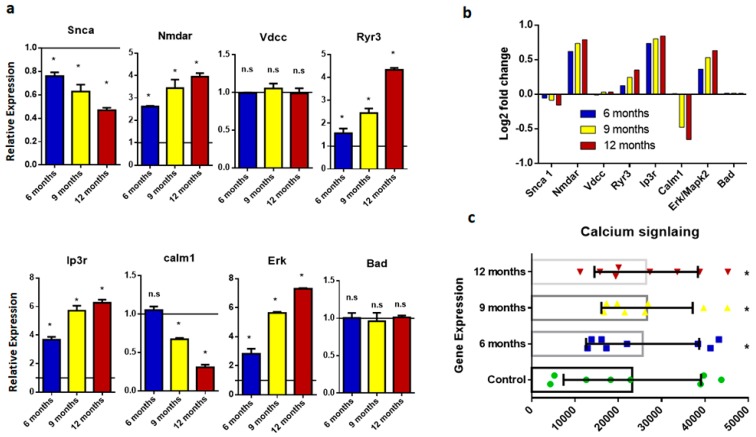
Expression analysis of genes involved in calcium signaling. (**a**) Histograms representing relative mRNA expression of *Snca*, *Ip3r*, *Nmdar*, *calm1*, *Vdcc*, *Erk*, *Ryr3*, and *Bad*. Represented values are the mean and SD obtained from three individual experiments performed for each sample. The line on the x-axis at 1.0 represents the gene expression of control samples. (**b**) Bar plots representing fold-change values represented in the log2 scale showing comparative gene expression in the brain of six-, nine-, and 12-month-old samples. Baseline shows expression in the control sample. (**c**) Scattered-dot plot showing expression of genes involved in calcium signaling. * Expression is significantly different from WT (*p* ≤ 0.05). n.s., Nonsignificant.

**Figure 5 ijms-20-01219-f005:**
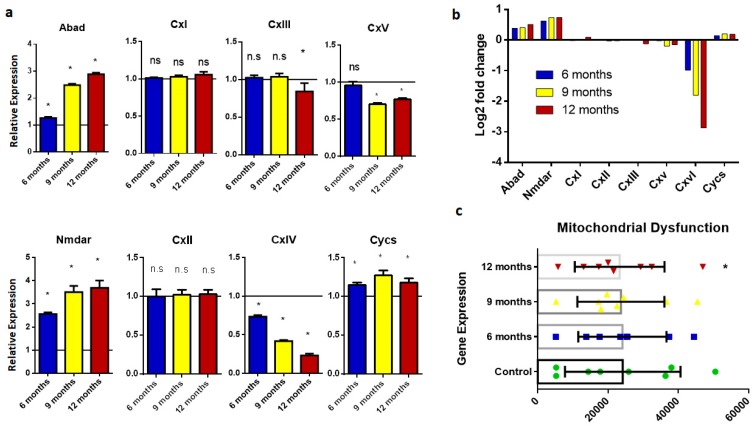
Expression analysis of genes involved in mitochondrial disruption. (**a**) Histograms representing relative m RNA expression of *Abad*, *Nmdar*, *Cx i–v*, and *Cycs*. Represented values are the mean and SD obtained from three individual experiments performed for each sample. The line on the x-axis at 1.0 represents the gene expression of control samples. (**b**) Bar plots representing fold-change values represented in the log2 scale showing comparative gene expression in the brain of six-, nine-, and 12-month-old samples. Baseline shows expression in control samples. (**c**) Scattered-dot plot showing expression of genes involved in mitochondrial dysfunction. * Expression is significantly different to WT (*p* ≤ 0.05). n.s., Nonsignificant.

**Figure 6 ijms-20-01219-f006:**
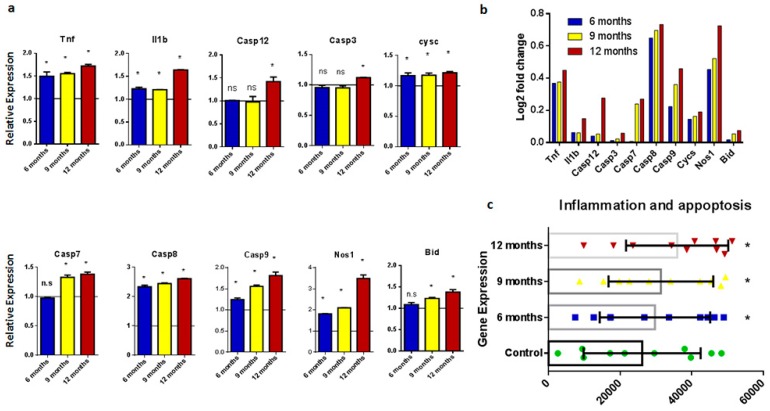
Expression analysis of genes involved in neuroinflammation and apoptosis. (**a**) Histograms representing relative mRNA expression of *Tnf*, *Il1b*, *Casp 3*, *Casp 7*, *Casp 8*, *Casp 9*, *Casp 12*, *Nos1*, *Cysc*, and *Bid*. Represented values are the mean and SD obtained from three individual experiments performed for each sample. The line on the x-axis at 1.0 represents the gene expression of control samples. (**b**) Bar plots representing fold-change values represented in the log2 scale showing comparative gene expression in the brain of six-, nine-, and 12-month-old samples. Baseline shows expression in control samples. (**c**) Scattered-dot plot showing expression of genes involved in calcium signaling. * Expression is significantly different to WT (*p* ≤ 0.05). n.s., Nonsignificant.

**Figure 7 ijms-20-01219-f007:**
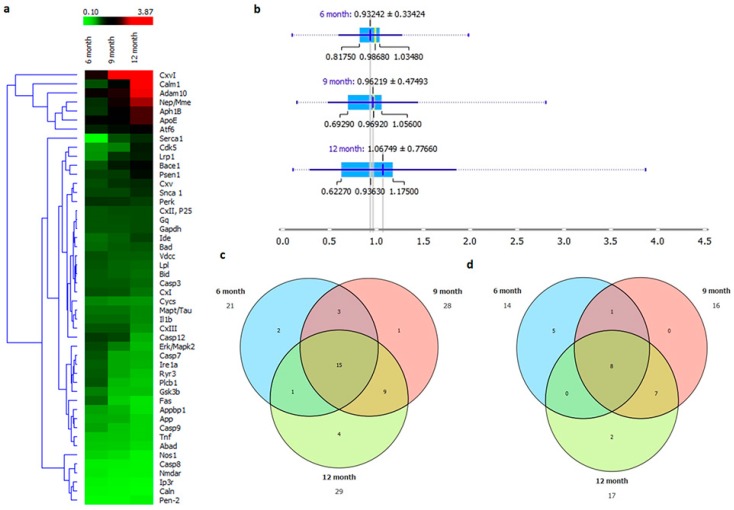
(**a**) Heat map showing comparison of all genes in this study expressed in diseased samples, divided into control, six-, nine-, and 12-month-old APP/PS1 transgenic mice. (**b**) Box plot showing comparison mean with SD of all genes involved in the onset of Alzheimer’s disease analyzed in three age groups. (**c**,**d**) Venn diagram representing the number of over- and underexpressed genes.

**Figure 8 ijms-20-01219-f008:**
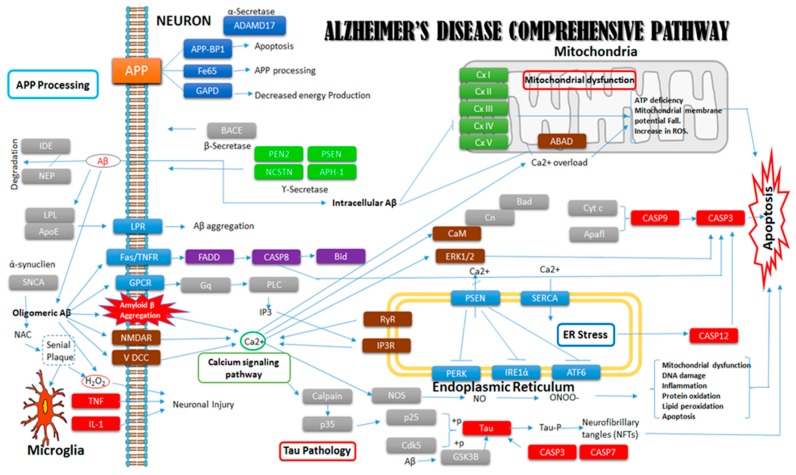
Comprehensive pathway of Alzheimer’s disease derived from the Kyoto Encyclopedia of Genes and Genomes (KEGG). This pathway shows a detailed summary of all the important genes involved in the molecular mechanisms in disease onset. Solid square boxes of different colors represent proteins while Colored squares without fill represents molecular events of the disease respectively.

**Table 1 ijms-20-01219-t001:** Primer sequence of genes used for expression analysis.

Gene Name	Reverse Primer Sequence 5′ to 3′	Forward Primer Sequence 5′ to 3′
*Adam10*	atcgaatcctgccatttcac	agccagagttgtgcgttttt
*Abad*	gacttccagcgggttatcaa	cagtgtcatgcccactatgc
*Aph1B*	gctgttcaggctcgcatatt	aatcaccatgaatgcccact
*Apo-E*	gtgctgttggtcacattgct	cagtgccgtcagttcttgtg
*App*	ggccctcgagaattacatca	gttcatgcgctcgtagatca
*Atf6*	ggccagactgttttgctctc	cccatacttctggtggcact
*Serca1*	tggccgatgataacttctcc	gagcccatcagtcaccaagt
*Bace1*	tttgtggagatggtggacaa	tacacaccctttcggaggtc
*Bad*	gggatggaggaggagcttag	cccaccaggactggataatg
*Bid*	tcacagacctgctggtgttc	gtctggcaatgttgtggatg
*Vdcc*	cgttctcatcctgctcaaca	tatgctcccaatgacgatga
*Calm1*	actgggtcagaacccaacag	gttctgccgcactgatgtaa
*Caln*	cagagggtgcttcgattctc	aaggcccacaaatacagcac
*Casp12*	ttcccaggaacagctgagtt	tcacgtggacaaagcttcag
*Casp3*	tgtcatctcgctctggtacg	tcccataaatgaccccttca
*Casp7*	tttgcttactccacggttcc	cacgggatctgcttcttctc
*Casp8*	ggcctccatctatgacctga	gcagaaagtctgcctcatcc
*Casp9*	aagaccatggctttgaggtg	aagtccctttcgcagaaaca
*Cdk5*	gtccatcgacatgtggtcag	acgacgttcaccaaggatgt
*P25*	cgtccactagtgagctgctg	cccacctcagaggagatgac
*Cxi*	ctctccccagtaccctcgac	gggagtgggcctgaaattag
*Cxii*	cctttgggaaccacagctaa	tcaaagttcccaggaagcag
*Cxiii*	gttcgcagtcatagccacag	tagggccgcgataataaatg
*Cxiv*	gtgtccccactgatgaggag	cagccaaaaccagatgacag
*Cxv*	gaaactggaccaggtggaga	gataccctgggtgttgccta
*Cycs*	gggaggcaagcataagactg	tctgccctttctcccttctt
*Perk*	tggtgactgctatggaccaa	gttccatctgggtgctgaat
*Fadd*	acaatgtggggagagactgg	aggtcagccaccagattcag
*Fas*	ttgcaagacatgtcggaaag	cctgcatggcagttacacac
*Appbp1*	gcagccagggaagatactca	tcttctccgctgaccagatt
*Gapdh*	aagggctcatgaccacagtc	acacattgggggtaggaaca
*Gq*	cacgctcaagatcccataca	ggctacacggtccaagtcat
*Nmdar*	cagcaggactggtcacagaa	tttgttccccaagagtttgc
*Gsk3b*	gaggagagcccaatgtttca	aatttgctcccttgttggtg
*Ide*	gaggcgttccaaaaacacat	gacagccaacatttcctggt
*Il1b*	gaccttccaggatgaggaca	tccattgaggtggagagctt
*Ip3r*	gaatttccttcgttgccaaa	cgatgcagttctggttctca
*Ire1a*	cccaaatgtgatccgctact	agaatgttgtggggcttcag
*Lpl*	ttttctgggactgaggatgg	gtcaggccagctgaagtagg
*Lrp1*	gacagcaaacgaggcctaag	acaggggttggtcacttcag
*Erk*	tccttttgagcaccagacct	agcagatgtggtcattgctg
*Mapt*	gtggccaggtggaagtaaaa	gtggagatgtgtccccagac
*Ncstn*	ctgaccactctggctccttc	gctgctgaagttggttcctc
*Nep*	aggcggacaacctctactca	cgaggctggtcaaaatgaat
*Nos1*	agcacctaccagctcaagga	atagtgatggccgacctgag
*Plcb1*	catccaggaggtggttcagt	ccctttcatggcttcctgta
*Psen1*	cctcatggccctggtattta	tcagccatattcaccaacca
*Pen-2*	cgggtatccaatgaggagaa	gcgagaatgatcacccagaa
*Ryr3*	gtgcagcctctactcccttg	atgtcctccaccttgtctgg
*Snca*	ggagtgacaacagtggctga	caggcatgtcttccaggatt
*Tnf*	cgtcagccgatttgctatct	cggactccgcaaagtctaag

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
