# Peer review of "Comparative Gene-Expression Analysis of Alzheimer’s Disease Progression with Aging in Transgenic Mouse Model"

_ijms, 2019, doi:10.3390/ijms20051219_

Reviewer 1 Report

In this paper, the authors performed an expression profiling of the most important genes involved in Alzheimer’s disease in an APP/PS1 transgenic mouse model. I think that the paper is very good since it covered the most dysregulated pathways in AD, associating these genes with disease progression. The obtained results are quite interesting and the methodologies are appropriate. The manuscript has a good rationale, but there are some errors and suggestions to be done:

Major concerns include:

- Is it possible to include the names of genes instead of the number in the Venn diagram?

- I suggest to put a summary table with the most significantly differentially expressed genes for each investigated pathway, together with their function in AD pathogenesis.

- Has a dysregulation of these genes also been found in other neurological disorders?

- In my opinion, the table 3 is not very clear. I suggest the authors to simplify it, if it’s possible.

Minor concerns include:

Language needs a careful revision. Here are listed some mistakes:

- Line 172: please delete one point

- Line 268: please correct “om” in the text

- Line 269: “Venn” instead of Wenn

- Figure 7: d panel is missing in the figure

- Line 275: please insert a point in the sentence

Author Response

Response to Reviewer 1:

Comments and Suggestions for Authors

Comments: In this paper, the authors performed an expression profiling of the most important genes involved in Alzheimer’s disease in an APP/PS1 transgenic mouse model. I think that the paper is very good since it covered the most dysregulated pathways in AD, associating these genes with disease progression. The obtained results are quite interesting and the methodologies are appropriate. The manuscript has a good rationale, but there are some errors and suggestions to be done:

Response: We are really thankful for a scholarly review of our article and giving us encouraging remarks. All the scientific comments and suggestions given by honorable reviewer have been considered and amendments have been done accordingly.

Major concerns include:

Comments: - Is it possible to include the names of genes instead of the number in the Venn diagram?

Response: We are really thankful for the suggestion given by respected reviewer. Because of low space in the venn diagram its not possible to mention the genes in that diagram so I have added a table in supplementary file (S Table 1) mentioning the up regulated and down regulated genes witnessed in this study. I hope the reviewer will consider it satisfactory.

Comments- I suggest to put a summary table with the most significantly differentially expressed genes for each investigated pathway, together with their function in AD pathogenesis.

Response: As honorable reviewer suggested a table of the genes along with its function. A detailed table has been added in the supplementary file (S Table 2) with brief information of the genes as well as its function. 

Comments- Has a dysregulation of these genes also been found in other neurological disorders?

Response: Respected reviewer questioned about the involvement of mentioned genes in other neurological disorders as well. Obviously few of important genes which leads to neuroinflammation and apoptosis are part of pathological pathways of other diseases like Parkinson’s, Huntington disease etc. As this study tries to cover Alzheimer’s disease only that’s why the role of these genes in other disease have not been addressed.

Comments- In my opinion, the table 3 is not very clear. I suggest the authors to simplify it, if it’s possible.

Response: Amendments have been done according to the instructions of honorable reviewer over all in the manuscript and in figures where necessary.

Minor concerns include:

Comment: Language needs a careful revision. Here are listed some mistakes:

- Line 172: please delete one point

- Line 268: please correct “om” in the text

- Line 269: “Venn” instead of Wenn

- Figure 7: d panel is missing in the figure

- Line 275: please insert a point in the sentence

Response: We are really thankful for a respected reviewer for such a detailed review and there mentions for typographical error and mistakes. All the above mentioned errors have been amended and discrepancies have been corrected according to the instruction of reviewer.

            After all the changes and improvements according to recommendations of reviewer, we are hopeful that reviewer and editors will find this revised manuscript satisfactory and eligible for acceptance. 

Reviewer 2 Report

The manuscript entitled “Comparative Gene Expression Analysis of Alzheimer’s disease progression with aging in transgenic mouse model” is interesting research to reveal the AD progression of gene expressions with aging. The authors showed that gene expression profiling of the major molecular events with aging. However, I have several major comments.

In Introduction, the authors described “The main purpose of the present study is to devise a cost effective, efficient and comprehensive method to dissect the molecular basis of AD” in line 81. However, they did not describe how this result contribute to devise cost effective, efficient and comprehensive.

Furthermore, they will study the use of next-generation sequencing and microarrays to cover a broader spectrum of genes with AD. After all, these methods are high cost. It may be misunderstanding for readers that there is little point in results of this study. Please describe these points in discussion and conclusion.

In Materials and Methods, they described “Expression of different genes involved in pathology of AD were determined in the brain using end point RT-PCR in 5, 9 and 12-month-old transgenic and control mice by using primers” in line 118. These are shown in line 91 and 324. However, in results, their results showed as 5months in figure 4b and 6b, but they showed 6 months in other figures.

Which is right, 5 or 6? The purpose of this study is gene expressions with aging, so this difference may be very important.

In Results, why are controls showed only “controls”? Are the control mice no different with age?

Please describe the methods and results in detail.

It is hard for readers to see figures, because description between “a” and “b” is not standardized. Some figures are tandem, but others are transverse.

Maybe the color is wrong in figure 6. In figure 5b, Cxv and Cxvi is wrong. In figure7, “d” is lack.

Furthermore, Cx3 looks significantly higher than controls in figure 5a but looks lower in figure 5b.

In Discussion, they describe that Mitochondrial complexes I, II and III showed no significant changes compared to wild type in line 347, but in figure 5a, the expression of Cx3 looks significantly higher than controls. Please revise this point.

Author Response

Response to Reviewer 2:

Comments and Suggestions for Authors

Comments: The manuscript entitled “Comparative Gene Expression Analysis of Alzheimer’s disease progression with aging in transgenic mouse model” is interesting research to reveal the AD progression of gene expressions with aging. The authors showed that gene expression profiling of the major molecular events with aging. However, I have several major comments.

Response: We really feel honored that our respected reviewer has invested his precious time to review our manuscript. His scientific comments and remarks are taken critically. We are thankful for remarks and suggestions given by the reviewer which will help us in improving our manuscript.  

Comments: In Introduction, the authors described “The main purpose of the present study is to devise a cost effective, efficient and comprehensive method to dissect the molecular basis of AD” in line 81. However, they did not describe how this result contribute to devise cost effective, efficient and comprehensive. Furthermore, they will study the use of next-generation sequencing and microarrays to cover a broader spectrum of genes with AD. After all, these methods are high cost. It may be misunderstanding for readers that there is little point in results of this study. Please describe these points in discussion and conclusion.

Response:  As mentioned by the respected reviewer that the manuscript lacks the information on how this study will help in designing an effective, efficient and cost-effective method to dissect the molecular basis of AD. Along with it, he mentioned that the use of costly tools like next-generation sequencing and microarray is confusing. We would like to mention that finding of this study provides a birds-eye view of the pathological pathways and molecular events on this disease so the data from more sensitive techniques like next generation sequencing and Microarray with a huge set of genes will provide us will a set of most important genes to be the focus in one assay. The final goal of our whole project is to make a diagnostic tool which should be efficient accurate and not economically expensive and in access of common man. In future plans, we want to utilize human brain samples as well as cerebrospinal fluid for analysis. The points mentioned by respected reviewer have been addressed in the revised manuscript. I hope that the reviewer will find is acceptable.

Comments: In Materials and Methods, they described “Expression of different genes involved in pathology of AD were determined in the brain using end point RT-PCR in 5, 9 and 12-month-old transgenic and control mice by using primers” in line 118. These are shown in line 91 and 324. However, in results, their results showed as 5months in figure 4b and 6b, but they showed 6 months in other figures. Which is right, 5 or 6? The purpose of this study is gene expressions with aging, so this difference may be very important.

Response: We are thankful for the reviewer to mention the typographical error in the manuscript. It is actually for 6 months. So manuscript has been thoroughly gone through and this typo have been amended as recommended by the honorable reviewer.

Comments: In Results, why are controls showed only “controls”? Are the control mice no different with age?

Response:  Respected reviewer has asked about the control mice used in this experiment. As the main aspect of this study is to analyze gene expression with aging. So in this study 3 age groups were considered 6, 9 and 12 months of age. So mice of these 3 age groups from transgenic as well as the wild type were used in this study. So the age of control mice is the same as its corresponding transgenic disease model mice. We have added the description in materials and methods to avoid any ambiguity.

Comment: It is hard for readers to see figures because description between “a” and “b” is not standardized. Some figures are tandem, but others are transverse. Maybe the color is wrong in figure 6. In figure 5b, Cxv and Cxvi is wrong. In figure7, “d” is lack.

Response: As directed by respected reviewer Figures have to be considered again and tried to do some improvement and the description of the figures have been amended as well. I hope that the revised version of the manuscript should be better and the description will be more understandable for readers. I hope reviewers and editors will accept our changes.

Comment: Furthermore, Cx3 looks significantly higher than controls in figure 5a but looks lower in figure 5b.In Discussion, they describe that Mitochondrial complexes I, II and III showed no significant changes compared to wild type in line 347, but in figure 5a, the expression of Cx3 looks significantly higher than controls. Please revise this point.

Response: Respected reviewer has mentioned the expression of Cx3 which differs from the description in the manuscript. Samples for Cx3 have been re-processed and results have been revisited and the mentioned in the revised manuscript.

All the changes and improvements have done according to the instructions and recommendations of our reviewers so I hope that reviewers and editors will find this revised manuscript satisfactory and will be considered for acceptance for publication.

Round  2

Reviewer 2 Report

The authors conscientiously answered reviewers' comments. However, figure 4 and 5 have mistaken yet. Please recheck absolutely.

Author Response

Round 2 Response to Reviewer 2:

Comments and Suggestions for Authors

Comments: The authors conscientiously answered reviewers' comments. However, figure 4 and 5 have mistaken yet. Please recheck absolutely.

We are really thankful from the core of our heart to honorable reviewer for an encouraging review to our manuscript. As indicated by reviewer we have replaced the figures with improved one in the revised manuscript. More over the whole manuscript have been re-read and corrections have been done where necessary. The revised manuscript has been uploaded with track change option to detect the corrections. 

All the changes and improvements have done according to the instructions and recommendations of our reviewers so I hope that reviewers and editors will find this revised manuscript satisfactory and will be considered for acceptance for publication.